# PCR Test Positivity and Viral Loads during Three SARS-CoV-2 Viral Waves in Mumbai, India

**DOI:** 10.3390/biomedicines11071939

**Published:** 2023-07-08

**Authors:** Chaitali Nikam, Wilson Suraweera, Sze Hang (Hana) Fu, Patrick E. Brown, Nico Nagelkerke, Prabhat Jha

**Affiliations:** 1Thyrocare Laboratories, Navi Mumbai 400703, India; chaitali.nikam@thyrocare.com; 2Centre for Global Health Research, Unity Health Toronto and Dalla Lana School of Public Health, The University of Toronto, Toronto, ON M5B 1W8, Canada; wilson.suraweera@unityhealth.to (W.S.); hana.fu@unityhealth.to (S.H.F.); patrick.brown@utoronto.ca (P.E.B.); niconagelkerke@gmail.com (N.N.)

**Keywords:** SARS-CoV-2 pandemic, PCR positivity, Aleph Delta Omicron waves, Mumbai, India

## Abstract

SARS-CoV-2 polymerase chain reaction (PCR) tests generally report only binary (positive or negative) outcomes. Quantitative PCR tests can provide epidemiological information on viral transmission patterns in populations. SARS-CoV-2 transmission patterns during India’s SARS-CoV-2 viral waves remain largely undocumented. We analyzed 2.7 million real-time PCR testing records collected in Mumbai, a bellwether for other Indian cities. We used the inverse of cycle threshold (Ct) values to determine the community-level viral load. We quantified wave-specific differences by age, sex, and slum population density. Overall, PCR positivity was 3.4% during non-outbreak periods, rising to 23.2% and 42.8% during the original (June–November 2020) and Omicron waves (January 2022), respectively, but was a surprisingly low 9.9% during the Delta wave (March–June 2021; which had the largest increase in COVID deaths). The community-level median Ct values fell and rose ~7–14 days prior to PCR positivity rates. Viral loads were four-fold higher during the Delta and Omicron waves than during non-outbreak months. The Delta wave had high viral loads at older ages, in women, and in areas of higher slum density. During the Omicron wave, differences in viral load by sex and slum density had disappeared, but older adults continued to show a higher viral load. Mumbai’s viral waves had markedly high viral loads representing an early signal of the pandemic trajectory. Ct values are practicable monitoring tools.

## 1. Introduction

Various waves of SARS-CoV-2 virus infection in India have led to one of the world’s highest totals of confirmed COVID-19 cases (45 million as of 1 October 2022), with about 3–4 million excess deaths in independent studies analyzing registered deaths or national surveys. By contrast, less than 0.6 million deaths are reported by official government data sources [1,2].

Three major viral waves occurred in India. The first wave of six months occurred from June 2020 to November 2020 and peaked in late September 2020, with moderate increases in excess deaths from the virus [1]. We refer to this wave as “Aleph”. The Aleph wave was mostly due to the original virus that originated from Wuhan Province, China, including variants B.1.1.7 (Alpha) and B.1.617.1 (Kappa). The Delta wave of four months occurred from March to June 2021 and was predominantly due to the B.1.617.2 variant known for greater transmissibility and reduced responsiveness to vaccines [3,4], and was characterized by the largest spike in excess deaths. The Omicron wave, mostly in January 2022, caused widespread infection but far lower mortality and was predominately due to the B.1.1.529 variant.

Despite India collecting large amounts of testing data in central repositories, little testing or viral load data are available publicly. In the absence of reasonably representative national data, careful examination of sub-national settings can inform our understanding of the virus and transmission patterns in India [5]. Intergenerational households are the norm in Mumbai City, with a population of 13 million, about 65% of whom live in cramped, often poorly ventilated slum housing, allowing for easy transmission of the virus [6]. Moreover, Mumbai is considered a bellwether setting which showed the earliest increases and declines during each viral wave [7]. The Municipal Corporation of Greater Mumbai (MCGM), the responsible local government, reports over one million real-time polymerase chain reaction (PCR) confirmed cases and 17,000 confirmed COVID deaths since March 2020 [8,9]. However, these reports underestimate the true numbers as, even by the end of the first wave, serosurveys suggested that about a third of adults, including nearly half of adult slum dwellers, had SARS-CoV-2 antibodies [7,10].

Here, we examine data from a private but widely-used laboratory in Mumbai, capturing data from over 2.7 million individual PCR tests, with a subset providing information on viral load, as defined by the inverse of the cycle threshold (Ct) values, indicating how many times the machine needed to copy the virus’s genetic material before detecting that material. Ct values are considered a proxy for viral load, particularly if tested in the early stages of the infection [11]. We quantify differences in positivity and viral load during the three waves and quantify each wave by sex, age, and slum population density, as well as the time lags.

## 2. Methods

### 2.1. Data Sources

We included 2.7 million individuals of all ages who underwent PCR testing between 9 April 2020 and 30 January 2022 by a large commercial laboratory (Thyrocare) with many franchises throughout Mumbai. Thyrocare has been approved by the Indian government for COVID-19 testing and conducts all testing in one facility in Navi Mumbai, with samples sent daily from franchise locations [12]. Participants had to pay INR 600–750 (USD 8 to 10) per test. All records were anonymized and had no identifiable patient information. The authors had no access to information that could identify individual participants during or after the data collection. The available data include the date of the test, age, sex, and six-digit postal code of tested individuals. Reasons for testing were not recorded but tested individuals include those referred by physicians, hospitals, quarantine centers, and self-referrals from both public and private sectors. For each record, we extracted the PCR results and, from 22 August 2020 onwards, the Ct values among all PCR positives. We mapped individual postal code areas to municipal wards (Appendix A). Individuals from about 5% of postal code areas that cut across municipal wards were assigned to the ward with the largest population in that postal code. We collected publicly available PCR-confirmed cases and deaths, published in tabular form by age, sex, and municipal ward, from the MCGM COVID-19 portal [9,13,14]. MCGM provides estimates of the sex-specific total population and proportion living in slums (Appendix A) for each ward.

### 2.2. Laboratory Methods

Nasopharyngeal, oropharyngeal, or combined specimens from each person were collected into a barcoded sterile tube containing a transport medium and transported within one day to the central Thyrocare laboratory. All six commercial PCR kits used for diagnosis were approved by the Indian Council of Medical Research (ICMR; Appendix A), with independent validation to establish each assay’s high sensitivity and specificity. The Ct value was defined as the number of cycles required for the fluorescent signal to cross the detection threshold. Ct levels are inversely proportional to the amount of target nucleic acid in the sample (i.e., the lower the Ct level, the greater the amount of target nucleic acid; each unit of Ct represents a doubling of viral load). Ct values between 0 and 35 were considered test positive, and values < 30 were considered strongly positive [15,16,17,18].

### 2.3. Statistical Methods

Usually, laboratory PCR test results only provide binary outcomes, i.e., positive or negative. However, test results contain a great deal more information, specifically Ct values. Although there are challenges to relying on single Ct values for individual-level decision-making, even a limited aggregation of Ct value data may predict the trajectory of the pandemic in a population [19]. We calculated the daily PCR positivity rate (percent positive among total tested) and median Ct values of the Thyrocare data. We plotted the seven-day moving average of community-level PCR positivity rates and median Ct values from Thyrocare data together with official PCR confirmed cases, and death counts from MCGM’s COVID-19 dashboard. We used Pearson correlation coefficients to determine the simple associations between these time series. 

We used generalized additive models [20,21] to model temporal changes in the daily median Ct value and the PCR positivity rate over time. We assumed the daily median Ct values to be normally distributed and applied a logistic model for the number of daily positive/negative PCR tests. We included a day-of-week effect and a smoothly varying time trend by including a second derivative penalty. The model included a daily (normally distributed) overdispersion term. Model parameters were estimated by maximum likelihood. 

The age distributions of MCGM confirmed cases and Thyrocare cases were slightly different between the outbreak periods (Appendix A); thus, we adjusted the ages to the five-yearly age distribution of the 2011 census [22] in all calculations of PCR positivity rates and median Ct values.

Given that the PCR positivity rate and the median Ct values varied by age, sex, and municipal wards over the three pandemic waves (Appendix A), we applied a multivariate logistic model to investigate the effects of age (10-year classes with 20–29 years as the reference), sex (males as the reference), and the slums population density (high: more than 60% of the population living in slums; medium: 33–60%; and low: up to 33% with low slum density as the reference) on PCR positivity rate for each wave. 

We applied a similar proportional logistic model with the same explanatory variables to examine high versus medium viral load groups for each viral wave by classifying individuals into Ct quartile range groups (high: 25% or less; medium: 25–75%; low: 75% or more) and then comparing the highest viral load (i.e., low Ct) to the medium range as the reference. Comparisons of low Ct to the high Ct value quartiles yielded similar results. We used R software version 4.1.0 to fit the generalized additive models and SAS software version 9.4 to perform all other statistical analyses [23,24].

### 2.4. Ethics Statement

The study used anonymized data with no personal identifiers. Nonetheless, the study is covered by the ethical approval provided under an overall REB approval for the use of anonymized data (REB #15-231) by Unity Health Toronto. 

## 3. Results

Over 2.7 million PCR tests for SARS-CoV-2 were conducted at Thyrocare between April 2020 and January 2022, of which 2.2 million were conducted during the non-outbreak months. The overall PCR positivity rate was 5.3%, ranging from 3.4% during the non-outbreak months but rising to 23.2% and 42.8% during the Aleph and Omicron waves, respectively. However, a surprisingly small increase to 9.9% occurred during the Delta wave. PCR positivity rates were generally higher in women than men and higher at older ages and in areas with higher slum density (Table 1).

The median Ct value for the period from September 2020 to January 2022 was 24.0 (interquartile range 19.0–28.0). Median Ct values were similar during the non-outbreak months (25.0) and the Aleph wave (26.0) but notably lower during the Delta (23.0) and Omicron waves (23.2). This difference of about two Ct units represents an approximately four-fold higher viral load. During the Aleph and Omicron waves, the median Ct values were similar in areas with low, medium, or high slum density, but during the Delta wave, medium and high slum density areas had two or three units lower Ct values (or 4-fold to 8-fold higher viral load).

The PCR testing data covered all 24 wards and 83 of 91 postal codes within Mumbai and Mumbai Suburban districts in the MCGM area (Appendix A). The age distributions of men and women testing PCR positive in the Thyrocare laboratory data were similar to the overall distributions among 863,000 confirmed cases (case numbers in both sexes were 232,000 of Aleph, 377,000 of Delta, and 254,000 of Omicron), a subset of data reported by the MCGM [14,25] (Figure 1). In both Thyrocare and MCGM data, the Aleph wave was characterized by a dual peak of PCR positivity around age 30 and 60, whereas the PCR positivity rate during the Delta and Omicron waves peaked at around age 30. 

The seven-day averages of MCGM reported PCR-confirmed cases correlated temporally with Thyrocare PCR positivity rates and the inverse of the median Ct values (Appendix A; with higher viral load shown to rise vertically). The Pearson correlation coefficients of temporal agreement (association) between PCR confirmed case counts in MCGM, and both PCR positivity rates and median Ct values for each wave were: Aleph (r = 0.46, n.a.), Delta (r = 0.94, −0.67), and Omicron (r = 0.85, −0.88), respectively. Daily deaths, which are likely substantially undercounted, were also correlated with Thyrocare results. 

The rise in median viral load (inverse of Ct values) was substantial during the Delta wave, with a six-unit drop in Ct value, which is a 64-fold increase in community viral load. By contrast, during the Omicron wave, the viral load rose less sharply with an absolute Ct value difference of approximately three, that is, an eight-fold higher viral load, and the duration of the higher viral load was shorter than during the Delta wave. The predicted smoothed daily time series of PCR positivity rates and median Ct values by our generalized additive models (Figure 2) were similar to their seven-day daily median counterparts (Appendix A). The 95% confidence intervals of the daily predicted values show greater uncertainty for the daily median Ct value from September 2020 to February 2021 and for both the daily median Ct value and PCR positivity rates from November to December 2022, largely reflecting lower testing volumes during this period.

The beginning, peak, and end of each COVID wave were determined from the rise and fall of individual curves of viral load (Ct values) and PCR positivity rates using the seven-day averages, as shown in Appendix A. In each wave, the first detectable change was in the viral load, followed by the PCR positivity rate, and lastly by MCGM-reported deaths (Appendix A). During the Delta wave, the increase in viral load began on 1 February 2021 and peaked on 31 March 2021, preceded by about 5 and 17 days of the rise and peak in PCR positivity rate, respectively. During the Omicron wave, the increase in viral load (11 December 2021) and the peak (2 January 2022) also preceded the rise in cases and peak PCR positivity rate by about 10 days. For the beginning-to-peak duration, the Delta wave lasted twice as long (46–58 days) as the Omicron wave (20–22 days). For the peak-to-end duration for the Delta wave, viral load increases lasted for 49 days, PCR positivity rates for 72 days, MCGM-reported confirmed cases for 90 days, and MCGM-reported deaths lasted the longest with 99 days.

Because of the strong age, sex, and slum density patterns of both PCR positivity rates and median Ct values, multivariate analyses considering each one separately are required. Among the Thyrocare tested population, the PCR positivity rate was higher in females than males in all three viral waves (Figure 3), and there was a notable age gradient in the adjusted odds ratio of being PCR positive when compared to the reference age group of 20–29 years (age test for trend for each has *p* < 0.0001; Appendix A). During the early Aleph and Delta waves, areas with high or medium slum density had higher odds of PCR positivity than low slum areas, but these differences were not seen during the Omicron wave. 

Figure 4 shows a similar examination of the multivariate predictors of viral load, comparing the highest quartile to the middle two quartiles during the Delta and Omicron waves, which correspond to an absolute difference of 5 and 3.7 in Ct values (or 32-fold and 13-fold differences in community viral load). Females tended to have higher viral load than males during the Delta wave but not during the Omicron wave. Similarly, areas with high or medium slum density had higher odds ratios of high viral load during the Delta wave, but these differences disappeared during the Omicron wave. During the Delta wave, ages 30 years or older had somewhat higher odds ratios of high viral load, but during the Omicron wave, the higher odds ratios were seen at ages 60 years and older. 

## 4. Discussion and Conclusions

Our analyses of a large dataset of PCR-positivity rates and viral load in Mumbai, India, show the remarkably high levels of transmission that occurred during the Delta and Omicron waves. The Delta wave was characterized by rapid, 64-fold increases in population medians of viral load compared to the weeks prior to that wave. This rise in viral load preceded changes in PCR-confirmed positivity rates by at least seven days in both the Delta and Omicron waves, reflecting increases in transmission caused by these high viral loads. The Delta wave was characterized by high viral loads in areas of high or medium slum density and women. By contrast, such differences in viral load by sex and slum density were not perceived during the Omicron wave. This may reflect the lower pathogenicity of the Omicron strain in combination with (unobserved) changes in reasons for being tested. While all three waves showed rising PCR positivity rates with age, a notably increased risk of high viral load persisted among adults over the age of 60 years during the Omicron wave. 

India’s Delta wave was particularly lethal, causing nearly 3 million deaths [2]. This high total arose from the biological features of the virus itself, including tropism for infecting lower respiratory tract cells that would directly increase case fatality rates. The major excess in Delta wave deaths may also reflect widespread community transmission, in particular by multiple exposures to infected people, as well as the probability of becoming infected with the most infectious strain. The rapid 64-fold increase in viral load during the Delta wave was consistent with rapid and mostly uncontrolled community transmission, including widespread intergenerational household transmission. Paradoxically, the much smaller increase in PCR test positivity rates during the Delta wave (only about 10% versus 3% during non-outbreak weeks) than seen in other waves may reflect such widespread multi-generational infection within homes, so many infected individuals simply did not get tested. An alternative explanation might be that Delta was not highly transmissible but extremely pathogenic. 

Women had higher PCR positivity during each of the waves, but the viral load difference was seen only during the Delta wave and not during the Omicron wave. This finding and the disappearance of differences by slum population density between the Delta and Omicron waves likely reflects widespread infection during the Delta wave and the rapid uptake of SARS-CoV-2 vaccination in Mumbai (Appendix A), which, like India’s overall vaccine rollout, aimed to reach slum populations quickly [26]. If this is the correct explanation, it offers some reassurance against possible future waves. Some caution is needed, however, as the highest viral load during the Omicron wave was seen in the oldest age groups (above 60 years), suggesting a reduced hybrid immunity (from a combination of infection and vaccination) [27]. Our finding lends an additional argument that follow-up vaccine boosters for the elderly are needed in India in view of the reports of the emergence of Omicron variant B 2.75 and the fact that the Delta wave occurred several months ago. Unfortunately, India only has high coverage of two-dose vaccination regimes [28,29,30].

Our study adds to others, suggesting that the use of serial trends in Ct values, even in non-representative populations, is a robust method to track viral load, particularly in low- and middle-income countries where PCR testing access is limited. Most Ct values studies are from high-income countries, where transmission patterns differ substantially from the intergenerational transmission commonly seen in low- and middle-income countries (LMIC). In our review of the literature, we identified 67 Ct studies, of which only six were in LMICs, including two from India. None explored determinants of viral load as we do. Despite this strength, our study also has some obvious limitations. First, we did not have data on why people got tested in these private laboratories. Subtle changes in who is tested may bias the use of PCR positivity rates for mapping trajectories of the infection [31], and we cannot exclude that such biases also affected the Ct values. However, changes in Ct values and their value as an early signal were so dramatic that we believe that this signal, representing higher viral loads at the community level, was far stronger than the inherent noise in Ct values [32,33]. Similarly, various PCR machines were used, and this might affect the trends in observed Ct values. However, the ICMR and others have established the high overall sensitivity and specificity of each machine independently. Finally, the study represents only one large urban setting in India, but Mumbai has been reflective of COVID-19 patterns for urban India from the origin of the pandemic [7,34].

Our study has several other strengths as well, notably a large sample size with testing procedures being mostly uniform over time. Moreover, the geographic and age distribution of the Thyrocare tested population resembled confirmed cases reported by the MCGM. 

Future studies that apply Ct values should stratify populations by past infection or vaccination status, as these events might have an important bearing on the relationship between infection and viral load [35]. Routine access to Ct values collected by governments may also provide a more robust early warning system to track future pandemic waves of SARS-CoV-2. 

## Figures and Tables

**Figure 1 biomedicines-11-01939-f001:**
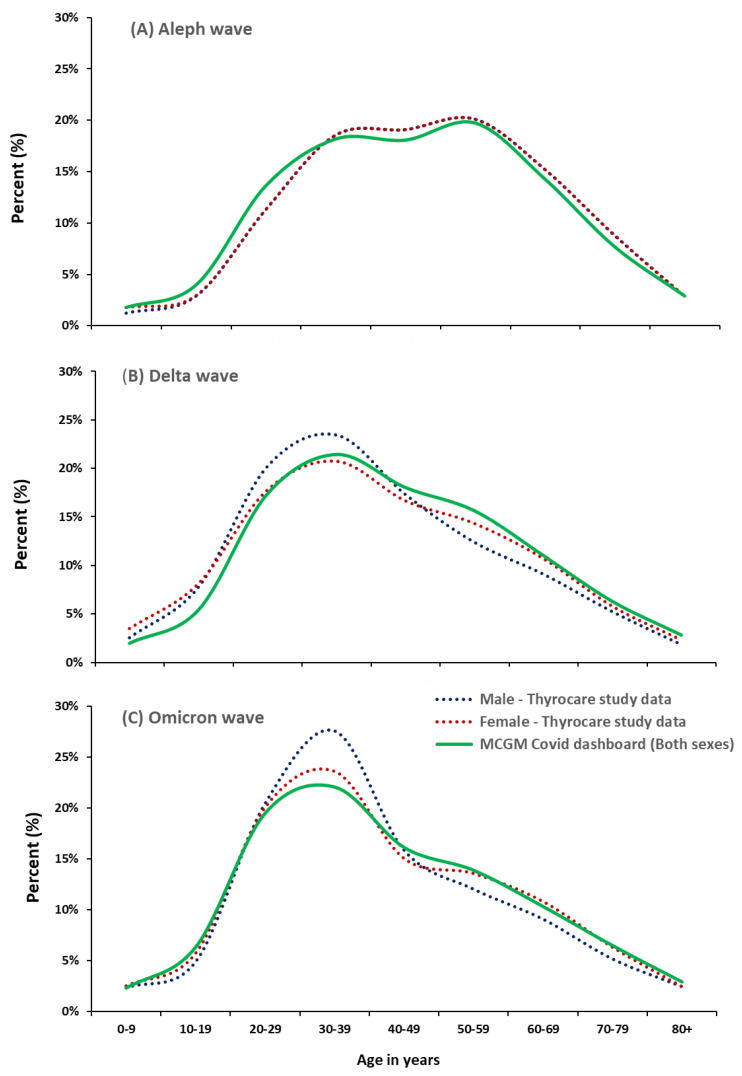
Age distribution of PCR-confirmed cases between MCGM COVID dashboard (both sexes in green) and Thyrocare study data (in dotted red: females, blue: males) during the three viral waves. (**A**) Aleph wave (June 2020–November 2020), (**B**) Delta wave (March 2021–June 2021), (**C**) Omicron wave (January 2022). *p*-values of the Kruskal–Wallis similarity test for age distributions of both sexes for (**A**–**C**) were 0.53, 0.31, and 0.73, respectively.

**Figure 2 biomedicines-11-01939-f002:**
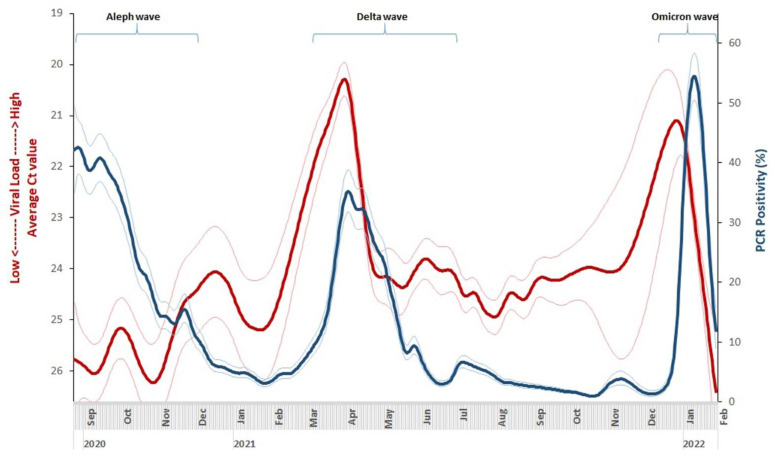
Trends in smoothed daily viral loads (inverse of cycle thresholds) and polymerase chain reaction (PCR) positivity in Thyrocare laboratory data in Mumbai, India. The trend graphs apply generalized additive models to examine changes in the daily median Ct value and PCR positivity rates over time. The thick red line represents the smoothed daily Ct values, and the surrounding pink lines are the corresponding 95% confidence interval limits; the thick dark blue line represents the smoothed daily PCR positivity, and the surrounding light blue lines are the corresponding 95% confidence interval limits. Daily Ct values are plotted on the reverse y-axis on the left, with lower Ct values corresponding to higher viral load. The 95% confidence intervals of the daily predicted values show greater uncertainty for daily median Ct values from September 2020 to February 2021 and also show greater uncertainty for both Ct daily median Ct value and PCR positivity from November to December 2022. The graph is restricted to the time period from 24 August 2020 to 30 January 2022, therefore only partially covering the Aleph wave, which occurred from June to November 2020.

**Figure 3 biomedicines-11-01939-f003:**
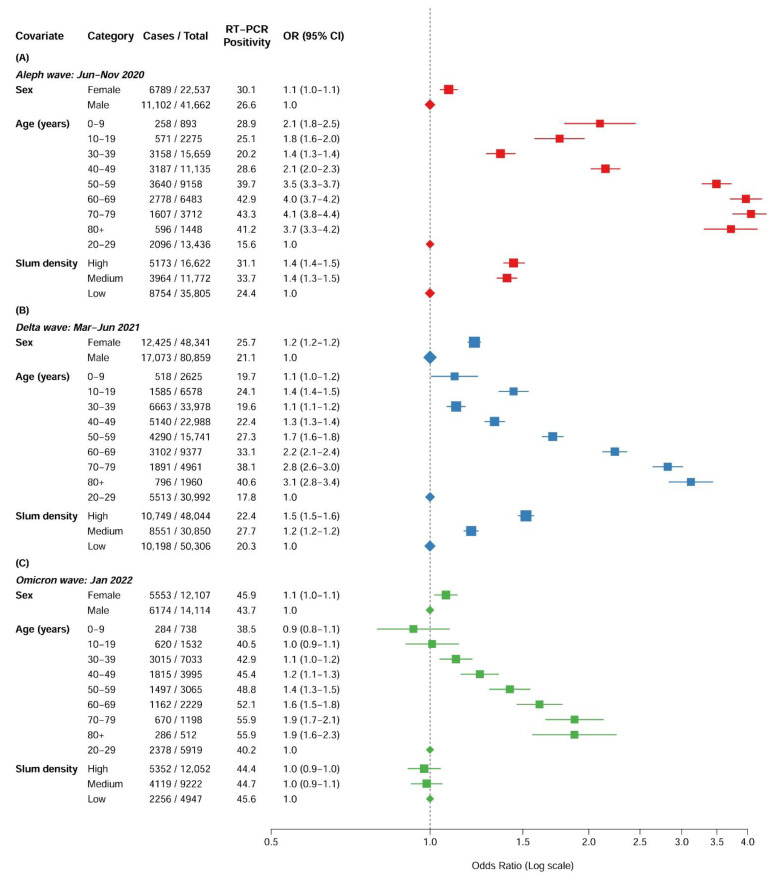
Adjusted odds ratios of PCR positivity at three viral outbreak periods in Thyrocare data. (**A**) Aleph wave: June–November 2020 (in red), (**B**) Delta wave: March–June 2021 (in blue), (**C**) Omicron wave: January 2022 (in green). For each outbreak period, a separate multivariable binary logistic model was fitted on the PCR positivity rate in our study data to assess the independent effects of covariates age (10 yearly groups), sex, and slum status of the residence ward. The age group of 20–29 years (which usually had the lowest-reporting PCR positivity among age groups), male sex, and areas of low slum density (that represent the higher social status of living) were selected as the respective reference groups.

**Figure 4 biomedicines-11-01939-f004:**
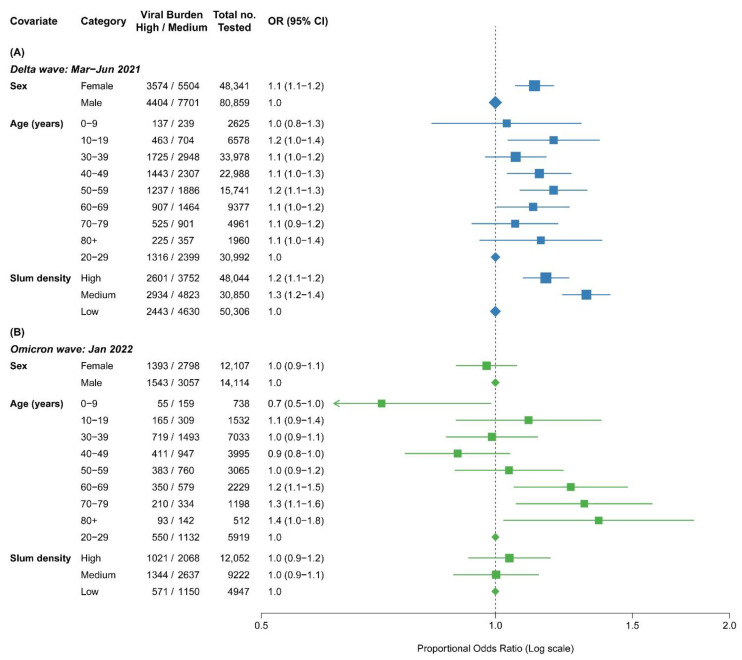
Proportional odds ratios of high (lowest 25% of Ct values) vs. medium (25–75% of Ct values) viral load groups. (**A**) Delta wave: March–June 2021 (in blue), (**B**) Omicron wave: January 2022 (in green). We fitted two separate multivariable proportional logistic models for the Delta wave and for the Omicron wave to assess the effects of age, sex, and slum density of the residence ward between viral load categories (distinguished by Ct value quartile ranges; high viral load: lowest 25% of Ct, medium: 25–75%, low: highest 75% of Ct). The median quartile range was chosen as the reference group. The related (25%, 50%, 75%) quartile values of each viral wave were: Delta wave: 18.0, 23.0, and 28.0, and Omicron wave: 19.5, 23.2, and 27.6. We assessed the effects of each independent covariate, including age (10 yearly groups), sex, and slum density of a residence ward on high viral loads vs. medium viral loads using the proportional odds ratios. The age group of 20–29 years usually had the lowest-reporting PCR positivity among age groups, male in sex, and areas of low slum density that represent higher social status of living were selected as the relevant reference groups.

**Table 1 biomedicines-11-01939-t001:** Summary of official case counts, community-level PCR positivity rates, and median Ct values from MCGM COVID-19 dashboard and Thyrocare-tested population during the pandemic period from April 2020 to January 2022.

Data Source/ Characteristics		Overall Period April 2020 to January 2022 (22 Months)	Pandemic Periods			
			Non-Outbreak Periods (11 Months)	Aleph Wave (January–November 2020) 6 Months	Delta Wave (March–January 2021) 4 Months	Omicron Wave (January 2022) 1 Month
Official PCR confirmed cases (MCGM dashboard)						
Total PCR positive		1,022,979	160,288	232,484	376,528	253,679
Case rate per 1000 population *		191.6	12.8	33.9	82.4	222.1
	Age < 40 years	130.5	7.6	18.1	54.3	159.1
	Age ≥ 40 years	334.9	24.8	71.0	148.4	369.9
	Female	n.a.	n.a.	29.0	76.1	191.6
	Male	n.a.	n.a.	38.1	87.8	248.1
	Low slum areas	261.2	16.4	45.1	123.5	262.5
	Medium slum areas	173.8	13.1	32.8	77.4	195.7
	High slums	151.5	8.2	25.9	68.8	164.3
Community-level PCR testing (Thyrocare data)						
	Total No. tested in 000	2717.3	2181.7	64.2	445.1	26.2
	Test positive in 000	155.0	78.8	17.9	46.6	11.7
Overall PCR Positivity (%) †						
	All ages	5.3	3.4	23.2	9.9 ‡	42.8
	Age < 40 years	5.1	3.4	18.8	9.5	41.3
	Age ≥ 40 years	6.5	3.9	36.9	11.7	49.2
	Female	5.75	3.75	25.87	11.46	44.06
	Male	4.98	3.2	21.91	8.98	41.82
	Low slum areas §	4.2	3.3	20.4	7.2	44.3
	Medium slum areas §	21.7	8.3	28.8	25.3	42.2
	High slum areas §	17.7	5.2	25.8	19.9	42.8
Median Ct value †						
	Overall	24.0	25.0	26.0	23.0	23.2
	Inter quartile range (Q1, Q3)	(19.0, 28.0)	(20.0, 28.0)	(21.0, 31.0)	(18.0, 28.0)	(19.5, 27.6)
	Age < 40 years	24.0	25.0	25.0	23.0	23.1
	Age ≥ 40 years	24.0	24.0	26.0	22.0	21.9
	Female	24.0	25	25.0	23.0	23.1
	Male	24.0	24	26.0	23.0	23.2
	Low slum areas §	24.0	25.0	26.0	24.0	22.9
	Medium slum areas §	22.3	23.0	25.0	21.0	23.0
	High slum areas §	23.0	23.1	26.0	22.0	23.4

Notes: 1. * Case rates were adjusted to the duration (number of months) of each pandemic wave, as shown in column label headings, to reflect the annualized rate. 2. † Crude PCR positivity per 100 tested and median Ct values were varying by age. For a comparison of PCR positivity rates between viral waves, we adjusted the ages of the tested population to the five-year age distribution of the census population in 2011. Details of Government COVID dashboard data, overall PCR positivity rate, and Ct values are shown in Appendix A Web Appendix. 3. ‡ The reported lower test positivity rate during the Delta wave from March to June 2021 was likely due to an increase in test frequency of healthy individuals for work and travel purposes following government requirements. 4. § Slum population density—high: more than 60% of the population living in slums, medium: 33% to 60%, and low: up to 33%.

## Data Availability

Data presented in this study are available upon request from the corresponding author.

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
