# Peer review of "PCR Test Positivity and Viral Loads during Three SARS-CoV-2 Viral Waves in Mumbai, India"

_biomedicines, 2023, doi:10.3390/biomedicines11071939_

Round 1
Reviewer 1 Report
The article entitled as "PCR test positivity and viral load during three SARS-CoV-2 viral waves in Mumbai, India" is an interesting work.
However, the article needs some improvements please see the comments below:
Line no. 36 to 38: We refer to this wave as “Aleph”. The Aleph wave was due mostly to the original virus that originated from Wuhan Province China, with some B.1.1.7 (Alpha) and B.1.617.1 (Kappa) variants of concern.
Kappa variant is not a Variant of concern, please amend accordingly.
Improve the introduction while talking more about the variants. The introduction can be used of this article (https://doi.org/10.1016/j.biopha.2022.113522).
In the manuscript you have mentioned:
Our study has several strengths also, notably a large sample size with testing procedures being mostly uniform over time. Moreover, the geographic and age distribution of the Thyrocare tested population resembled that of confirmed cases reported by the MCGM.
There must be some limitations? which must be highlighted.
Additionally, I suggest writing a conclusion separately and highlight your major findings.
The English is acceptable however, needs a check to avoid any potential, mistakes.
Author Response
Reviewer 1
The article entitled as "PCR test positivity and viral load during three SARS-CoV-2 viral waves in Mumbai, India" is an interesting work.
Reply: We thank the reviewer for this insightful comments and valuable suggestions.
However, the article needs some improvements please see the comments below:
- Line no. 36 to 38: We refer to this wave as “Aleph”. The Aleph wave was due mostly to the original virus that originated from Wuhan Province China, with some B.1.1.7 (Alpha) and B.1.617.1 (Kappa) variants of concern. Kappa variant is not a Variant of concern, please amend accordingly.
Reply: Thanks for this valuable comment. We revised the text in lines 36-38 by replacing “with some B.1.1.7 (Alpha) and B.1.617.1 (Kappa) variants of concern” with the following “including variants B.1.1.7 (Alpha) and B.1.617.1 (Kappa) “ to address this issue.
- Improve the introduction while talking more about the variants. The introduction can be used of this article (https://doi.org/10.1016/j.biopha.2022.113522).
Reply: Although we recognize the value of engaging in an in-depth discussion on variants, as suggested by the reviewer, we have concerns that such a digression would divert attention from our primary focus on the epidemic in Mumbai. Furthermore, regrettably, our expertise does not extend to this particular field.
- In the manuscript you have mentioned:
Our study has several strengths also, notably a large sample size with testing procedures being mostly uniform over time. Moreover, the geographic and age distribution of the Thyrocare tested population resembled that of confirmed cases reported by the MCGM. There must be some limitations? which must be highlighted.
Reply: The reviewer is correct that stating limitations is very important. However, the paragraph starting from line 312 lists both the limitations that we have recognized and the extent to which these might affect our findings.
- Additionally, I suggest writing a conclusion separately and highlight your major findings.
Reply: Following the reviewer's suggestion we have changed the title of the section “Discussion” into “Discussion and Conclusions” as our Discussion and Conclusions are hard to disentangle.

Reviewer 2 Report
The study interpreted the Ct values of SARS-CoV-2 PCR results between Apr20 and Jan22. The study might be useful for the colleagues in the field. The authors try to use a scientific angle with explanations to utilize the semi-quantitative values in analyzing/predicting the transmission patterns. The manuscript was well-written, adequate background information was provided, the results presented were clear and concise, the significance and implications of the findings were discussed and compared with other studies. Readers are alerted the limitations of the study. I have some comments in the current version of the manuscript so that the quality can be improved further.
- lines 52-53: I do not understand the role of MCGM. Is it a part of India government?
- The authors should present the data generated from the Thyrocare lab only. In order to avoid confusion, the data generated from MCGM should be avoided. If the authors want to compare their own data with the MCGM one, the authors can mention/discuss them in the main text and present the tables/figures in ‘Supporting Information’. The corresponding MCGM data in the tables/figures should be adjusted, then, the manuscript will be neater and tidier. For example, the green line, official PCR confirmed cases can be replaced by All (i.e. male + female) PCR confirmed cases.
- line 139: typo mistake, ‘January 2021’ should be ‘January 2022’
Author Response
Reviewer2
The study interpreted the Ct values of SARS-CoV-2 PCR results between Apr20 and Jan22. The study might be useful for the colleagues in the field. The authors try to use a scientific angle with explanations to utilize the semi-quantitative values in analyzing/predicting the transmission patterns. The manuscript was well-written, adequate background information was provided, the results presented were clear and concise, the significance and implications of the findings were discussed and compared with other studies. Readers are alerted the limitations of the study. I have some comments in the current version of the manuscript so that the quality can be improved further.
Reply: We thank the reviewer for insightful comments and useful suggestions.
- lines 52-53: I do not understand the role of MCGM. Is it a part of India government?
Reply: We added the phrase “the responsible local government” to line 53 to explain what MCGM is.
- The authors should present the data generated from the Thyrocare lab only. In order to avoid confusion, the data generated from MCGM should be avoided. If the authors want to compare their own data with the MCGM one, the authors can mention/discuss them in the main text and present the tables/figures in ‘Supporting Information’. The corresponding MCGM data in the tables/figures should be adjusted, then, the manuscript will be neater and tidier. For example, the green line, official PCR confirmed cases can be replaced by All (i.e. male + female) PCR confirmed cases.
Reply: We have carefully considered the reviewer's suggestions about the Tables and Graphs. We very much agree with the reviewer that Tables and Graphs are often imperfect compromises between the amount of information presented and clarity. While the suggested changes to our Tables would certainly improve clarity, it would remove information (specifically the agreement between Thyrocare and MCGM official figures) that we consider important and that the Table is supposed to bring across. After some experimentation, we therefore decided to keep the Table as it is. We have adjusted the legend of Figure 1 taking the reviewer's suggestions into account.
- line 139: typo mistake, ‘January 2021’ should be ‘January 2022’
Reply: January 2021. Oops. We are grateful to the reviewer for having spotted this. Now corrected.

Reviewer 3 Report
Authors examined data from a private but widely-used laboratory in Mumbai, 58 capturing over 2.7 million individual PCR testing data with a subset providing 59 information on viral load, as defined by inverse of the cycle threshold (Ct) values. Ct 60 values are considered as a proxy for viral load, particularly if tested in early stages of the 61 infection. Authors quantified differences in positivity and viral load during the three waves, 62 and quantify each wave by sex, age, and slum population density, as well as the time lags.
Although this manuscript is potentially interesting and useful, several issues arise.
1. Figure 1: Addition of total number may be useful.
2. The “Ct” value should be explained.
3. Was the antigen of SARS-CoV-2 measured? Was only PCR methods performed to diagnose the SARS-CoV-2 infection?
4. Was the same PCR test for SARS-CoV-2 performed among Aleph, Delta and Omicron waves?
5. Abstract conclusion: Was the second conclusion obtained from the results?
6. Mortality and incidence of thrombosis may be helpful.
Author Response
Reviewer 3
Authors examined data from a private but widely-used laboratory in Mumbai, capturing over 2.7 million individual PCR testing data with a subset providing information on viral load, as defined by inverse of the cycle threshold (Ct) values. Ct values are considered as a proxy for viral load, particularly if tested in early stages of the infection. Authors quantified differences in positivity and viral load during the three waves, and quantify each wave by sex, age, and slum population density, as well as the time lags.
Reply: We thank the reviewer for highly insightful and specific and valuable suggestions.
Although this manuscript is potentially interesting and useful, several issues arise.
- Figure 1: Addition of total number may be useful.
Reply: Thanks, Addition total numbers were now included into the figure 1 caption.
- The “Ct” value should be explained.
Reply: Good point. A short explanation is now included into the “Introduction”. Also further details are available in “Laboratory methods” section and references 15-18 would be useful for readers to have further information.
- Was the antigen of SARS-CoV-2 measured? Was only PCR methods performed to diagnose the SARS-CoV-2 infection?
Reply: Antigen tests for SARS-CoV-2 generally exhibit lower sensitivity compared to RT-PCR. Consequently, our study exclusively concentrated on the laboratory PCR method.
- Was the same PCR test for SARS-CoV-2 performed among Aleph, Delta and Omicron waves?
Reply: Yes the same PCR test but used 6 different test kits in randomly, all approved by the Indian Council of Medical Research for PCR testing purposes and the ranges of sensitivity >95% and specificity >99%. Details provided in supplementary Appendix S3.
- Abstract conclusion: Was the second conclusion obtained from the results?
Reply: Very sharp! Good point. Yes, we should have avoided “conclusions” that have not established directly from our analysis. We thus removed this “conclusion”
- Mortality and incidence of thrombosis may be helpful.
Reply: We very much agree that mortality and incidence of thrombosis play vital roles in understanding the epidemic. However, unfortunately thrombosis data are not available in a form that it can be aligned with our study data.
